# MINISTAR to STARLITE: Evolution of a Miniaturized Prototype for Testing Attitude Sensors

**DOI:** 10.3390/s25175360

**Published:** 2025-08-29

**Authors:** Vanni Nardino, Cristian Baccani, Massimo Ceccherini, Massimo Cecchi, Francesco Focardi, Enrico Franci, Donatella Guzzi, Fabrizio Manna, Vasco Milli, Jacopo Pini, Lorenzo Salvadori, Valentina Raimondi

**Affiliations:** 1Institute of Applied Physics “Nello Carrara”—National Research Council, 50019 Sesto Fiorentino, Italy; v.nardino@ifac.cnr.it (V.N.); v.raimondi@ifac.cnr.it (V.R.); 2GESTIONE SILO S.R.L., 50018 Scandicci, Italy; baccani.c@silo.it (C.B.); milli.v@silo.it (V.M.); pini.j@silo.it (J.P.); 3PROMEL COSTRUZIONI S.R.L., 50023 Impruneta, Italy; massimo.ceccherini@promel.it (M.C.); massimo.cecchi@promel.it (M.C.); fabrizio.manna@promel.it (F.M.); 4SAITEC S.R.L., 50063 Figline Valdarno, Italy; francesco.focardi@saitecsrl.com (F.F.); enrico.franci@saitecsrl.com (E.F.); lorenzo.salvadori@saitecsrl.com (L.S.)

**Keywords:** star tracker, Optical Ground Support Equipment (OGSE), star field simulation, optical stimulator

## Abstract

**Highlights:**

**What are the main findings?**
This paper describes a compact electro-optical system designed to generate synthetic star fields in apparent motion for realistic ground-based testing of star trackers.Working principles of the system and capability of the first prototype are described: simulation of stars and other celestial bodies, user-defined objects, and disturbance sources.

**What is the implication of the main finding?**
Early advancements are reported, enabling the transition from the initial MINISTAR prototype to the next-generation STARLITE system.Initial steps for instrument validation, characterization, and transition to a commercial model, and early results are presented.

**Abstract:**

Star trackers are critical electro-optical devices used for satellite attitude determination, typically tested using Optical Ground Support Equipment (OGSE). Within the POR FESR 2014–2020 program (funded by Regione Toscana), we developed MINISTAR, a compact electro-optical prototype designed to generate synthetic star fields in apparent motion for realistic ground-based testing of star trackers. MINISTAR supports simultaneous testing of up to three units, assessing optical, electronic, and on-board software performance. Its reduced size and weight allow for direct integration on the satellite platform, enabling testing in assembled configurations. The system can simulate bright celestial bodies (Sun, Earth, Moon), user-defined objects, and disturbances such as cosmic rays and stray light. Radiometric and geometric calibrations were successfully validated in laboratory conditions. Under the PR FESR TOSCANA 2021–2027 initiative (also funded by Regione Toscana), the concept was further developed into STARLITE (STAR tracker LIght Test Equipment), a next-generation OGSE with a higher Technology Readiness Level (TRL). Based largely on commercial off-the-shelf (COTS) components, STARLITE targets commercial maturity and enhanced functionality, meeting the increasing demand for compact, high-fidelity OGSE systems for pre-launch verification of attitude sensors. This paper describes the working principles of a generic system, as well as its main characteristics and the early advancements enabling the transition from the initial MINISTAR prototype to the next-generation STARLITE system.

## 1. Introduction

A satellite platform in space requires precise attitude control, which involves determining its orientation relative to a fixed reference frame. The primary inertial reference in space is defined by the so-called Fixed Stars on the Celestial Sphere, whose apparent positions remain effectively constant, rendering parallax effects negligible.

Star trackers are the most accurate pointing reference for satellites, reaching a precision in the order of the arcsec [1]. These optical instruments determine the spacecraft’s attitude (i.e., rotation axis orientation and corresponding angular velocity) by observing the positions and apparent motions of stars within their field of view (FOV). The observed stellar patterns are compared against an internal star catalog to estimate the instantaneous orientation of the spacecraft, taking into account the geometric alignment between the star tracker and the spacecraft body frame.

A star tracker is composed of an optical head equipped with a baffle, which observes the portion of the Celestial Sphere within its FOV; a detector located at the focal plane, used to acquire images of the star field; and an internal processing unit providing the image analysis and returning the absolute orientation of the sensor. The typical output is a four-element quaternion, easily converted into a direction cosine matrix or Tait–Bryan angles [2].

As a critical component of satellite missions, star trackers are typically deployed in redundant configurations to ensure resilience against failures and to maintain continuous attitude determination. Redundancy is especially important during satellite maneuvers, when temporary blinding may occur due to celestial bodies or stray light sources.

Given their critical role throughout the satellite’s operational lifespan, star trackers undergo rigorous ground testing to verify performance under space-relevant conditions. The initial testing phase involves separate verification of key subsystems, including optical parameters (e.g., focal length, aberrations, and distortion), detector and proximity electronics functionality, and the onboard image processing algorithms used to interpret star field data. These tests are typically carried out using Electronic Ground Support Equipment (EGSE), which replicates operational scenarios and provides diagnostic feedback for the system under test [3]. However, this verification phase is not capable of detecting malfunctions related to the integrated system (such as misalignment between optical elements and the detector, or damage incurred during assembly). To overcome these limitations, Optical Ground Support Equipment (OGSE) is required.

OGSE systems are laboratory-based setups specifically designed for testing fully assembled star trackers prior to their integration onto the satellite platform [4,5,6]. These systems perform tests by simulating a dynamic star field scene in relative motion with respect to the star tracker, which is presented directly to the optical head. The resulting attitude computed by the star tracker is then compared against the known reference used to generate the simulated scene, allowing assessment of the system’s accuracy. A primary limitation of this method is its incompatibility with post-integration testing, as it cannot be applied once the star tracker is mounted onto the spacecraft. This constraint motivates the development of a comprehensive, end-to-end testing solution capable of validating star tracker functionality both in laboratory conditions and after integration, including during pre-launch phases. To enable such capability, the test system must be miniaturized, with low mass and compact dimensions, to permit direct installation on the star tracker’s optical head, typically on or within its optical baffle.

A new generation of compact optical stimulation systems for star tracker testing has recently become commercially available. Notable examples include the STOS (Star Tracker Optical Stimulation for Sensors) developed by Airbus Space Equipment [7], the Optical Sky Stimulator (OSI) designed for the Jena-Optronik ASTRO APS star sensor [8], and the Terma Dynamic Optical Ground Support Equipment [4]. These systems generate dynamic simulations of star fields through the use of an optical projection system providing collimated beams (to match the star tracker focal) that present the simulated image in apparent motion to the sensor under test.

In addition to stars, these systems are capable of simulating various non-stellar objects and perturbations, such as stray light or artifacts induced by charged particle interactions. They can operate in an open-loop mode for sensor verification and robustness assessment or in real-time closed-loop configurations to evaluate star tracker behavior under realistic dynamic conditions. These simulators are designed to enable comprehensive, end-to-end testing of star trackers under representative operational environments.

Within the framework of the POR FESR 2014–2020 program (funded by Regione Toscana), the MINISTAR system was developed, a miniaturized electro-optical prototype designed to generate synthetic star fields in apparent motion, enabling realistic ground-based testing of star trackers [9,10]. MINISTAR supports the simultaneous testing of up to three units, allowing comprehensive evaluation of their optical, electronic, and on-board software performance. Given its reduced dimensions and mass, the system is suitable for direct integration onto the satellite platform, thereby enabling functional testing in the fully assembled configuration.

The system is capable of simulating star fields, as well as bright celestial bodies (e.g., Sun, Earth, Moon), user-defined objects, and various environmental disturbances, such as cosmic rays and stray light. Radiometric and geometric calibrations of the prototype have been successfully validated under controlled laboratory conditions.

Building on this experience, the ongoing PR FESR TOSCANA 2021–2027 initiative (also funded by Regione Toscana) has supported the development of STARLITE (STAR tracker LIght Test Equipment), a next-generation Optical Ground Support Equipment (OGSE) system. The system addresses the growing need for compact, high-performance OGSE solutions for pre-launch verification of spaceborne attitude sensors. Designed with a higher TRL, STARLITE leverages commercial off-the-shelf (COTS) components to achieve enhanced functionality and reliability while targeting commercial deployment. Specifically, STARLITE effectively aims to overcome the major challenges encountered during the development of its predecessor, MINISTAR. These include the technical challenges associated with the design and mechanical integration of a custom optical system, as well as the limited resolution of the MINISTAR display. In its final design, STARLITE will employ a compact commercial objective and a high-resolution display with enhanced luminance compared to its predecessor. The star positioning accuracy will be further improved through the adoption of enhanced calibration procedures. An additional enhancement is STARLITE’s increased compactness and the improved ease of mounting onto the star tracker’s baffle. The adoption of COTS components also contributes to a reduction in overall design and development costs.

The development of the instrument has been carried out through a collaborative effort involving a group of Tuscan partners, including SAITEC S.R.L. (project leader), GESTIONE SILO S.R.L., PROMEL COSTRUZIONI S.R.L., and the Institute of Applied Physics “Nello Carrara” of the National Research Council (IFAC-CNR). SAITEC is a high-tech company located near Florence with expertise in electronic and software systems, particularly in the development of Electrical Ground Support Equipment (EGSE), and a strong focus on the aerospace and avionics sectors. GESTIONE SILO contributes specialized capabilities in the design and fabrication of high-precision optical components and instruments. PROMEL COSTRUZIONI brings extensive experience in precision mechanics, including mechanical design and the production of high-precision parts for industrial and engineering applications. IFAC-CNR, a research institute based in Florence, plays a key role in applied physics research, with recognized competence in optics, photonics, and sensing technologies, bridging scientific research and industrial innovation.

This paper outlines the operating principles of a generic star tracker test system, detailing its main components and tracing the development path from the initial MINISTAR prototype to the more advanced STARLITE system. The discussion highlights key technical innovations and design optimizations that have driven this evolution. Experimental results obtained with MINISTAR, along with preliminary outcomes from STARLITE, are presented and analyzed.

## 2. Star Field Simulator Design Criteria

The design of a miniaturized dynamic star field simulator, such as MINISTAR or its successor STARLITE, is driven by a combination of optical, computational, and system-level requirements that shape both the hardware architecture and the underlying mathematical/software models. One of the primary design drivers is the need for fast and efficient computation of synthetic dynamic scenes. These scenes must incorporate not only stellar objects but also non-stellar sources (e.g., Sun, Moon, Earth) and disturbance phenomena such as stray light or cosmic radiation effects. A rough estimate of the computational demand per frame can be derived from timing constraints: to achieve realistic motion during a star tracker’s acquisition period, a sufficient number of simulation frames must be rendered. Given that typical star tracker acquisition rates are on the order of 10 Hz [5], the minimum required simulation frame rate—according to the Nyquist criterion—is 20 Hz to avoid aliasing and artifacts. However, to ensure proper synchronization and realistic motion perception, especially in the presence of frame-rate mismatches and potential latency, a significantly higher frame rate is desirable. Frame rates in the range of 50–90 Hz—comparable to common video standards—are technically preferable to support real-time, high-fidelity testing. Another critical requirement is the capability to test multiple star trackers simultaneously. This entails parallel rendering of coherent dynamic scenes for several optical heads, each with distinct orientations relative to the shared satellite reference frame. As a baseline, MINISTAR, as well as STARLITE, is designed to support simultaneous testing of at least three star trackers, necessitating synchronized scene generation across independent optical channels.

In terms of optical design, the simulator must provide a projected FOV large enough to match or exceed that of the star tracker under test. The star field image must completely fill the optical FOV and be in correct focus. As this type of star simulator targets compatibility with a wide range of star trackers, its optical and mechanical design must satisfy the following trade-offs:Sufficiently large projection pupil to illuminate, as fully as possible, the entrance pupil of the star tracker’s optical head, while maintaining compact and mechanically feasible optics;FOV equal to or greater than that of most commercially available star trackers;Parametric rendering model being capable of generating synthetic star fields within a configurable FOV and radiometric range, supporting user-defined magnitude distributions and custom object catalogs;A modular software architecture that incorporates an internal representation of a comprehensive star catalog (on the order of 10^5^ stars), enabling fast access, dynamic selection, and real-time rendering in the star tracker’s reference frame. This software is designed to be fully controllable via a graphical user interface for ease of use and test campaign configuration.

## 3. Model Description

The simulation model can be mathematically described as a procedure that generates a dynamic optical scene based on the following input parameters:The celestial coordinates (position) and apparent magnitudes of stars on the Celestial Sphere;The attitude of the simulated satellite platform, expressed either as a rotation versor or, equivalently, in quaternion form;The orientation of the star tracker’s optical axis (i.e., the direction of its field of view) relative to the satellite platform body frame.

The output of the simulation is a time-sequenced series of frames representing the apparent motion of the celestial scene as observed by each star tracker, assuming solidal motion with the satellite platform. Each frame spatially encodes the angular distribution of celestial objects—primarily stars—within the optical field of view of the star tracker, along with additional non-stellar elements such as planets and user-defined artifacts introduced to simulate disturbance effects (e.g., stray light, cosmic ray hits). Such angular distribution is rendered in two dimensions on a digital display surface and subsequently collimated through the optical projection system before being projected into the star tracker field of view.

### 3.1. Celestial Coordinate System

The reference frame adopted by the MINISTAR model is the one described by the celestial coordinates shown in Figure 1a. Declination and right ascension angles DEC and RA locate the position (more precisely, the *direction*) of each celestial object on the Celestial Sphere.

During the simulation, the satellite platform is modeled as rotating about a fixed axis at the center of the Celestial Sphere. As a result, each onboard star tracker observes a specific portion of the sky included in its FOV, determined by its fixed orientation relative to the platform’s body frame.

For each star tracker, the simulation process involves identifying the subset of stars and non-stellar objects that fall within its instantaneous field of view, given its orientation. These objects are then projected onto the tracker’s image plane, with their positions rotated and transformed to match how they appear in the star tracker’s local reference frame. The display image is subsequently projected into the optical path of the tracker.

### 3.2. Platform Orientation Angles

The orientation of the platform with respect to the celestial coordinates is expressed easily by the Tait–Bryan angles (a reformulation of Eulerian angles referred to a rotation around each Cartesian axis): ρ (roll, around x axis), π (pitch, around y axis), and ι (yaw, around z axis). The angles and their sign conventions are shown in Figure 1b, together with their relative positions with respect to the celestial coordinates axis system (c).

### 3.3. Star Tracker’s Optical Head Orientation Angles

The determination of the viewing direction for each star tracker’s optical head (co-rotating and integral with the satellite platform) requires the composition of two rotation matrices: the rotation matrix representing the satellite platform’s attitude with respect to the inertial reference frame (i.e., the Celestial Sphere) and the rotation matrix defining the fixed orientation of each optical head with respect to the platform (i.e., relative to the satellite body frame). By composing these two matrices, the absolute orientation of the star tracker’s optical axis in the celestial reference frame is obtained. This transformation is essential to determine the specific portion of the sky visible within the star tracker’s field of view at each simulation time step.

$R_{sat}$, the platform rotation matrix, is expressed as the product of three orthogonal rotation matrices with respect to the angles ρ, π, and ι  (the first rotation applied is the one by a roll angle ρ, then the pitch π, and then the yaw ι):(1)$$R_{sat}\left(\iota ,\pi ,\rho \right)=R_{z}\left(\iota \right)R_{y}\left(\pi \right)R_{x}\left(\rho \right),$$The rotation matrix with respect to the Celestial Sphere for the i-th optical head $R_{h_{i}}$ (i=1,…,N) is determined by applying the rotation matrix $R_{sat}$ to the rotation matrix of the generic head with respect to the platform by the angles $\rho_{h_{i}},\;\pi_{h_{i}},\;and\;\iota_{h_{i}}$ (i.e., the roll, pitch, and yaw angles in the platform reference frame). The composition of rotation sequences brings to(2)$$R_{h_{i}}\left(\iota ,\pi ,\rho ,\iota_{h_{i}},\pi_{h_{i}},\rho_{h_{i}}\right)=R_{sat}\left(\iota ,\pi ,\rho \right)R\left(\iota_{h_{i}},\pi_{h_{i}},\rho_{h_{i}}\right).$$

Note that all matrices are orthogonal and the matrix product is non-commutative (it is possible to define a different order of application of the rotations $R_{z}$, $R_{y}$, and $R_{x}$ and re-define the rotation matrices).

The final (rotated) direction of view for each optical head is calculated by applying $R_{h_{i}}$ to the initial position in the celestial sphere reference frame.

The time evolution of the system is obtained by varying the orientation of the platform with the time (i.e., varying the angles ρ, π, and ι as a function of time) and keeping constant the angles of the i-th star tracker’s head $\rho_{h_{i}},\;\pi_{h_{i}},\;and\;\iota_{h_{i}}$, being built in with the platform.

By applying the rotation matrix $R_{h_{i}}$, which defines the orientation of the *i*-th star tracker’s optical head in the inertial (celestial) reference frame, the model determines the corresponding viewing direction. Conversely, by applying the inverse rotation, the positions of stars on the Celestial Sphere are transformed into the reference frame of the *i*-th star tracker. The stars included in the simulated scene’s FOV can eventually be shown (i.e., rendered in the 2D display and projected into the star tracker’s head optics). It is worth noting that the matrix inversion involved in this transformation is computationally efficient, as it reduces to a simple transpose operation resulting from the orthogonality property of rotation matrices.

### 3.4. Quaternions

The view direction can, alternatively, be identified by a quaternion, i.e., a four-dimension vector describing a rotation of a fixed angle around a given rotation axis (with respect to the celestial reference frame). The (orthogonal) rotation matrix corresponding to the unitary quaternion $q=[q_{w},\;q_{x},q_{y},q_{z}]$ is(3)$$\begin{array}{c} R=\\ \left[\begin{array}{ccc} 1-2\left(q_{y}^{2}+q_{z}^{2}\right) & 2\left(q_{x}q_{y}-q_{w}q_{z}\right) & 2\left(q_{w}q_{y}+q_{x}q_{z}\right)\\ 2\left(q_{x}q_{y}+q_{w}q_{z}\right) & 1-2\left(q_{x}^{2}+q_{z}^{2}\right) & 2\left(q_{y}q_{z}-q_{w}q_{x}\right)\\ 2\left(q_{x}q_{z}-q_{w}q_{y}\right) & 2\left(q_{w}q_{x}+q_{y}q_{z}\right) & 1-2\left(q_{x}^{2}+q_{y}^{2}\right) \end{array}\right] \end{array}$$

The quaternion can, in fact, be expressed as a function of its rotation angle α and the corresponding rotation axis with director cosines $\cos\beta_{x,y,z}$:(4)$$q=\left[\begin{array}{c} \begin{array}{c} q_{w}\\ q_{x}\\ q_{y} \end{array}\\ q_{z} \end{array}\right]=\left[\begin{array}{c} \begin{array}{c} \cos\frac{\alpha }{2}\\ \sin\frac{\alpha }{2}\cos\beta_{x}\\ \sin\frac{\alpha }{2}\cos\beta_{y} \end{array}\\ \sin\frac{\alpha }{2}\cos\beta_{z} \end{array}\right]$$

The time evolution of the quaternion q is realized by varying with the rotation angle α and the director cosines $\cos\beta_{x,y,z}$ and calculating the new position. The software implementation of the model allows the simulation to be performed by using either the Tait–Bryan angles or the quaternions and allows for easy conversion between the two nomenclatures.

### 3.5. Fast Star Catalog Processing

The star catalog used for star field simulation is the widely used HIPPARCOS [11,12,13]. The catalog reports the astronomical coordinates of 118,218 stars with precision in the order of the milliarcsec (i.e., a factor $10^{3}$ smaller than the one used by the most precise star trackers) and the corresponding magnitude. The variables of interest extracted from the catalog are, for each star, the visual (apparent) magnitude, the B-V magnitude (defined using the Johnson–Morgan system [14], classifying stars based on their spectrum and used for correction of the tested star tracker instrumental spectral response curve) and the declination and right ascension angles for positioning the star on the Celestial Sphere.

The most straightforward way of simulating the apparent motion of the star is by applying the inverse rotation ${R_{h_{i}}}^{-1}={R_{h_{i}}}^{T}$ to the Celestial Sphere, i.e., to any star (and non-star object). Mathematically, $R_{h_{i}}$ is the rotation matrix bringing the generic optical head in a new direction of view in the Celestial Sphere reference frame. This is analogous, for an observer in the reference frame of the star tracker’s optical head, to applying the inverse rotation ${R_{h_{i}}}^{-1}$ to the Celestial Sphere, bringing the new pointed scene into the FOV of the instrument. Regrettably, applying ${R_{h_{i}}}^{-1}$ to the entire star catalog (comprising approximately $10^{5}$ stars) is computationally redundant and inefficient.

To accelerate computation, several star selection strategies are employed, with spatial proximity serving as the primary criterion: upon loading the star catalog into memory, it is partitioned into sub-catalogs, each representing a distinct region of the sky. Adjacent regions are designed to partially overlap (see Figure 2), and each sub-catalog includes only those stars within a predefined angular radius from the region center. The center of each region is spaced out by nearby region centers by a distance equal to FOV along the RA, DEC axis, forming a mosaic (with overlapped tiles) on the Celestial Sphere. The spatial partitioning of the catalog is defined at load time such that adjacent regions overlap sufficiently to guarantee that the entire FOV is fully contained within the selected region, regardless of its orientation relative to the declination (DEC) and right ascension (RA) axes. Each sub-catalog contains only the stars located within a specified angular distance from its central direction and, for a given viewing direction, only the sub-catalog associated with the closest region center is accessed. The underlying assumption is that, for a fixed viewing direction, only the stars within the corresponding sub-catalog fall within the field of view of the star tracker’s optical head. Consequently, only these stars are subjected to the inverse rotation ${R_{h_{i}}}^{-1}$, rather than applying the transformation to the entire HIPPARCOS catalog.

A second selection criterion is based on the magnitude range of the stars and will be presented in the next section.

### 3.6. Star Radiometric Simulation

#### 3.6.1. Radiometric Conversion from Display Digital Number to Apparent Magnitude

In analogy with [15], the magnitude *m* is calculated using the following formula:(5)$$m-m_{ref}=-2.5\log\left(X/X_{ref}\right)$$
where $m_{ref}$ is the reference magnitude, and X and $X_{ref}$ are, respectively, the measured quantity and the corresponding reference quantity.

We first define the bolometric magnitude (also known as absolute bolometric magnitude) by setting the zero-point of the magnitude scale at $M_{ref}=0$, corresponding to an object with a total radiative luminosity of exactly $F_{ref}=3.0128\times 10^{28}$ W, which is the absolute bolometric magnitude of the star Vega.

The absolute bolometric magnitude M of a source with luminosity M (expressed in W) is therefore given by(6)$$M=M_{ref}-2.5\log\left(\frac{F}{F_{ref}}\right)=71.197425-2.5\log F$$

Similarly, we define the zero-point of the apparent bolometric magnitude scale by specifying that $m_{ref}=0$ corresponds to an irradiance (flux density) $E=2.518021002\times 10^{-8}$ W m^−2^, which is the flux received isotropically from a source with M=0 located at the standard distance of 10 parsecs (based on the IAU 2012 definition of the astronomical unit [16]).

The apparent bolometric magnitude m for a given irradiance E (in W·m^−2^) is thus(7)$$m=m_{ref}-2.5\log\left(\frac{E}{E_{ref}}\right)=-2.5\log E-18.99735$$

Being expressed using a logarithmic scale, only a restricted range of values for m can be represented on the finite dynamics of the display (as in RGB displays, the value range is between 0 and 255 values for each color channel).

Fixing at 0 the reference magnitude $m_{ref}$, we obtain the formula for the pixel value I between (i.e., the brighter value)(8)$$I=255\cdot 10^{-0.4m}$$

I ranges between its maximum value admitted by the display dynamics $I_{max}=I(m=m_{ref})=255$ and the minimum value $I_{min}=I(m=6)≅1$; consequently, the theoretical range for m is approximately 6 magnitude units.

A reasonable work hypothesis is that the generic star tracker detector has a linear response to the detected photons, so stars too faint to be detected (i.e., out of a reasonably small magnitude range) can be simply ignored in the simulation. This consideration allows the number of stars on the catalog to be drastically reduced by selecting only the one of the magnitude range of interest. This second filter allows the simulation to deal with a drastically reduced number of stars (typically on the order of tens).

In practice, the theoretical range for m is considerably reduced. Considering that the display luminance must be adjusted to match the expected star photon flux, a portion of the display’s dynamic range is reserved for fine-tuning this software-based radiometric calibration. In addition, (as shown in Section 3.7) the generic star, albeit an angular point-like source, is usefully represented as a sub-pixel-resolved barycenter of a multiple-pixel distribution. Allocating a portion of the total pixel’s dynamics to barycenter fine-tuning further reduces the dynamic range required for magnitude simulation. Due to these reasons, the magnitude range is restricted to approximately 4.5 magnitudes, consistent with the capabilities of similar simulators currently available on the market [7].

#### 3.6.2. Sensor-Dependent Spectral Response Calibration

When simulating the star radiance using a synthetic image provided by a display (in our case, a commercial RGB 2D display), a huge technical problem arises: the display is unable to replicate the exact stellar spectrum. Furthermore, the instrumental response of the star tracker depends on the optoelectronic characteristics of its sensor, so we need a method to convert, for a generic sensor, the simulated value (display pixel luminance) to match the corresponding simulated magnitude measured by the sensor under test. With this working assumption, we can expect such a conversion to be approximated by a polynomial relation, based on the values of the parameters given by the star catalog for each star: the generic visual magnitude $M_{V}$ and the B-V magnitude $M_{B-V}$ (the so-called “star color”). The “calibrated” value for the magnitude observed by the star tracker under test can be set by comparison with a simulated star of known magnitude. The actual SW model supports the use of a second-degree polynomial as a correction formula (for taking into account a non-linear response to the simulated spectrum).

Simulation in MINISTAR, as well as in STARLITE, is performed using a single RGB channel of the display (typically the red channel) to reduce the chromatic aberration in the collimator optical system.

### 3.7. Angular Accuracy

Incoming star rays can be considered perfectly collimated; i.e., their angular distribution approximates a Dirac delta function. Under the ideal assumption of a star tracker equipped with ideal focusing optics, a star would be imaged onto a single point on the detector. As a consequence, finer angular resolution would be pixel-limited (i.e., the detector’s pixel pitch becomes a fundamental limit to the achievable angular resolution). This inherent constraint on resolution [17] is typically addressed by deliberately avoiding a pixel-limited point spread function (PSF) on the image plane. In practice, star tracker optical systems are designed with controlled, symmetric (with respect to the optical axis) aberrations such as intentional defocusing, resulting in an expanded star spot. This transforms a point-like source into a multi-pixel intensity distribution on the detector. By computing the barycenter of this distribution, the system achieves sub-pixel resolution, thereby improving angular precision beyond the physical detector pitch.

The simulation model adopted both in MINISTAR and STARLITE must replicate the star image distribution with a precision comparable to that of the star tracker. A single-pixel representation of a star is, in fact, inadequate, as it fails to provide sufficient angular resolution. For example, a 2000 × 2000 pixel display with a 20° field of view would offer a coarse resolution of only 0.01°, insufficient for accurately simulating the angular separations between stars. To address this, MINISTAR employs a sub-pixel precision model that utilizes the barycenter of a multi-pixel star spot to enhance angular resolution. A key constraint is that the Point Spread Function (PSF) of the star tracker optics must encompass that of the simulated star, so the system under test must remain insensitive to whether the input is an ideally resolved, point-like star or a discrete pixel distribution rendered on the display.

The achievable sub-pixel precision is determined by both the display’s dynamic range (pixel values between 1 and 255) and the number of pixels used to represent the star spot. As shown in Figure 3, the star’s position is identified using the barycenter of a square, multiple-pixel matrix, whose summed intensity equals the brightness value assigned to that star. In the best-case scenario (maximum brightness N), the barycenter position can be adjusted in steps as small as 1/N. Conversely, for the faintest visible star (value = 1), the minimum barycenter displacement is 1/2 pixel. The maximum achievable sub-pixel offset range within this setup is 1/2 pixel. Figure 3 illustrates various 2 × 2 pixel distributions, demonstrating sub-pixel variations in the barycenter position.

Using a 3 × 3 pixel distribution can improve sub-pixel resolution further, but it also increases the star’s apparent angular size. In the current implementation, 2 × 2 matrices are preferred to minimize the risk of resolving the star image into a non-point-like object. As said in Section 3.6.1, allocating more of the display’s dynamic range to achieve finer sub-pixel positioning and mitigating coarse angular resolution in faint stars succeeds (together with star luminance tuning) in reducing the maximum theoretical magnitude dynamic range.

### 3.8. Model Software Implementation

The physical and mathematical model described in Section 3.1, Section 3.2, Section 3.3, Section 3.4, Section 3.5, Section 3.6 and Section 3.7 has been implemented in a dedicated software library developed in standard C++. The library is designed for integration with a simulation framework. The objects of the library are instantiated by the main software managing the test procedure and used during the simulation phase. The library generates the star coordinates for the single frame, and the main software deals with the scene rendering and the star tracker output monitoring. The library provides, for each star tracker’s optical head, the coordinates and values for each star and non-stellar object for each frame of the simulated scenario. Every object in the simulated scene is described by the following:A couple of continuous coordinates (to be expressed using sub-pixel resolution as described in Section 3.7), normalized to the scene FOV and expressed in the reference frame of the simulated optical head;A magnitude value expressed in display units 0 to 255.

For non-stellar, extended objects, either a scalar radius or a bitmap representation—scaled to a user-specified angular size—is additionally provided. This approach allows the simulation of both star-like objects (extracted from HIPPARCOS catalog or alternative user-defined catalogs) and spatially extended objects such as planetary bodies.

Dedicated classes within the library also implement the star selection logic detailed in Section 3.5, including the memory allocation and organization of the stars in sub-catalogs (see Figure 2). At runtime, the software dynamically selects the appropriate sub-catalog based on the direction of view for each simulation frame. This enables the real-time generation of dynamic scenes and their rendering on the display.

The algorithmic structure governing the generation of each frame in the dynamic simulation sequence is illustrated in Figure 4.

## 4. Prototype Description

### 4.1. System Architecture

MINISTAR, as well as STARLITE system architecture, is made up of the following sub-systems:System software running on an external workstation that calls the mathematical model and provides the rendering of each frame of the simulated scene on a graphic device.A 2D display used to visualize the simulated star field and celestial scene in apparent motion.Optics for the collimation of the light rays emitted by the display, for generating the image of the star field focused at infinity (optical collimator).

The working principle of the simulation system is shown in the diagrams of Figure 5. The positions of both stars (from the star catalog) and other user-defined celestial bodies in the Celestial Sphere reference frame are acquired by the system software. On the basis of the (simulated) satellite platform direction, angular velocity, and the relative position of each star tracker’s optical head, the workstation generates the real-time sequence of frames of the simulated scene for each star tracker’s optical head under test. Each scene is rendered on the system’s display. An optical collimator focuses at infinity the simulated scene for each optical head. The system allows the real-time comparison of the simulated attitude with the star tracker output data, both in open and closed-loop configuration (i.e., in which the output attitude is used as input for the next step of the simulation).

Table 1 summarizes the key parameters that informed the selection and sizing of the optoelectronic components used in the MINISTAR prototype. 

Table 2 presents the nominal design requirements established for the STARLITE system, serving as the basis for its ongoing development.

The development of the STARLITE prototype is currently underway. Particular attention has been devoted to the significant enhancements implemented during the transition from the initial MINISTAR prototype to the next-generation STARLITE validation system. A key focus of this development has been the improvement of both the FOV and angular resolution to enable more accurate testing of star tracker performance in satellite attitude determination. STARLITE effectively addresses the critical limitations of its predecessor, particularly the mechanical complexity associated with the custom optical system. This mechanical simplification facilitates easier testing, especially under pre-launch conditions when the star tracker is integrated onto the satellite platform.

The following sections describe the initial outcomes of system design, with a focus on the selection and specification of critical components.

### 4.2. Display

#### 4.2.1. MINISTAR Display

The display module employed in the MINISTAR prototype featured the following characteristics:Model: OLED WUXGA (RGB) by eMagin Corporation (Hopewell Junction, NY, USA);Active area dimensions: 11.75 × 18.66 mm;Pixel pitch: 9.6 µm;Resolution: 1920 × 1200;White Luminance (color mode) > 150 cd/m^2^;Refresh rate 85 Hz;Gray levels: 256.

One of the main limitations of this display was its aspect ratio: despite a nominal resolution of 1920 × 1200 pixels, only a square region of 1200 × 1200 pixels could be effectively utilized for dynamic scene simulation due to constraints imposed by the optical system and FOV uniformity requirements.

#### 4.2.2. STARLITE Display

Based on the experience gained from MINISTAR, the STARLITE system incorporates a higher-resolution, square-format microdisplay: the SY103WAM01 from SeeYa, with the following specifications:Model: 1.03”Micro-OLED SY103WAM01 (2560 × 2560 RGB) by SeeYA (Hefei, China);Active area dimensions: 18.432 mm × 18.432 mm/1.03” diagonal (26.07 mm);Pixel pitch: 7.2 μm;Resolution: 2560 × 2560;Luminance: 1800 cd/m^2^ typical;Frame rate: 60~90 HZ;Grey levels: 256.

### 4.3. Optical System

In both the MINISTAR and STARLITE systems, the optical assembly functions as a collimator, projecting the microdisplay image at optical infinity to match the star tracker’s focal requirements.

#### 4.3.1. MINISTAR Optical System

The optical system used in MINISTAR (Figure 6) was based on a fully custom optical and mechanical design. This approach provided complete control over system-specific optical aberrations, allowing the optical layout to be optimized in accordance with the simulator’s performance requirements. However, this customization came with significant drawbacks, including elevated development and manufacturing costs, the need for high-precision mechanical mounts to maintain stringent alignment tolerances, and limited flexibility in adapting the system to different field-of-view (FOV) configurations.

The following input parameters were considered during the MINISTAR optical system design phase:Field of View (FOV): 20° (±10°);Exit Pupil Diameter: 35 mm;Pixel Size: ~8 µm;Optical System Length: 187 mm;Reference Wavelength: 632.8 nm;Exit Pupil: 35 mm.

#### 4.3.2. STARLITE Optical System

A comprehensive optical study was carried out for the STARLITE system using Zemax OpticStudio. The analysis focused on a selected set of commercial off-the-shelf (COTS) optical components, with particular emphasis on aspheric and achromatic doublets, as well as high-quality photographic objectives. A detailed analysis of candidate lenses (limited to Edmund Optics components) is presented in Table 3 and illustrated in the performance charts of Figure 7. To achieve an FOV comparable to MINISTAR, lenses with a focal length f of 40 mm have been considered. Given the active area side length s of the SeeYa STARLITE microdisplay (18.432 mm) and its resolution (2560 × 2560 pixels), the theoretical pixel-limited FOV can be estimated as(9)$$FOV=2arctan\left(\frac{s}{2f}\right)$$

For a nominal FOV of ±25.9° (corresponding to a 40 mm focal length), the resulting angular resolution is approximately 0.01°/pixel (~0.17 mrad). However, the actual optical performance of most evaluated lenses did not meet this theoretical resolution limit (see Table 3), primarily due to residual aberrations.

Ultimately, a high-performance photographic lens was selected for the STARLITE system: the Kowa 35 mm objective LM35XC—see Figure 8. This component offered an optimal trade-off between optical performance, mechanical compactness, and cost-effectiveness. The adoption of a commercial objective introduces valuable system flexibility: by interchanging the objective, the FOV can be tailored to the specific requirements of different star tracker models. This modular approach significantly enhances STARLITE’s compatibility with diverse sensor architectures and broadens its operational applicability in test and validation campaigns. The STARLITE mechanical interface is designed to accommodate standard C-mount optics, enabling easy replacement of the objective lens and thus allowing adjustment of the simulated field of view to match different star tracker configurations.

## 5. Results

### 5.1. MINISTAR Performance Evaluation

The geometric and radiometric calibration procedures adopted for the MINISTAR prototype have been described in detail in [10]. This section provides a summary of the key results.

#### 5.1.1. Geometric Calibration of MINISTAR

In general, uniformly spaced pixels on the display do not correspond to an equally regular angular grid when projected through the collimator. This discrepancy arises from optical distortions introduced by the projection system. Consequently, a dedicated geometric calibration and correction procedure is required to accurately map display coordinates to their corresponding angular positions in the projected field.

Geometric calibration employed a high-resolution Canon SX60 HS camera by Canon Italia s.p.a. (Cernusco Sul Naviglio, MI, Italy) at an 85 mm equivalent focal length. The calibration followed a two-step process:Camera distortion assessment: A regular test pattern was imaged at full resolution (4608 × 3456 pixels). From known dimensions and distances, the angular resolution per pixel was calculated. The observed deviation from an ideal grid was found to be below MINISTAR’s angular uncertainty, confirming negligible distortion. Consequently, the image captured by the camera can be regarded as a distortion-free reference.Distortion mapping: A grid of regularly spaced points was displayed on MINISTAR and imaged using the distortion-free camera.

Two sets of coordinates were obtained: a set of input coordinates x, y and a set of distorted (by the optical system) coordinates x’, y’. The resulting distorted coordinates were mapped to ideal coordinates via fitted bivariate polynomial surfaces: x(x’, y’) and y(x’, y’). These mappings allow for the inverse correction of the optical distortions introduced by the MINISTAR display system, i.e., the coordinates x’(x,y), y’(x,y) to be set on the display corresponding to a pixel at the corrected coordinates x,y on the simulated scenario projected into the sensor under test (Figure 9a). The root mean square (RMS) error of the fit was found to be 0.51 pixels, confirming accurate geometric calibration. Figure 9b shows the measurement setup.

#### 5.1.2. Radiometric Calibration of MINISTAR

For a generic pixel in an image acquired by a camera observing a radiance source, the expected digital output, expressed in digital numbers (DN), can be modeled as(10)$$DN=k\tau \int_{\lambda_{0}}^{\lambda_{1}}S_{D}(\lambda )L\left(\lambda \right)d\lambda .$$
where $L\left(\lambda \right)$ is the (Lambertian) radiance (expressed as power per unit solid angle and per unit projected area), averaged in the spectral channel $\Delta \lambda =\lambda_{1}-\lambda_{0}$, $S_{D}(\lambda )$ is the spectral response of the detector, integrally normalized to units in the spectral interval $\left[\lambda_{0},\lambda_{1}\right]$ and null elsewhere, τ is the integration time, and k is a constant depending on the camera entrance pupil area, focal length, pixel dimensions, and optoelectronics parameters. In concise form, we can write(11)$$DN=k\tau {\langle L(\lambda )\rangle }_{\Delta \lambda }$$
with ${\langle \ldots \rangle }_{\Delta \lambda }=\int_{\Delta \lambda }S_{D}\left(\lambda \right)\ldots d\lambda$ indicating the spectral average operator.

The radiometric calibration of the MINISTAR display consists of determining the constant k for the calibration camera and radiometrically characterizing the spectrum of the generic pixel of MINISTAR display. For determining the value of k, the image of a radiometrically calibrated Lambertian source HL-3P-INT-CAL has been acquired with a DALSA 1M60 camera mounting a NIKON 45 mm objective, set to infinity, as detailed in [10]. In the hypothesis of Lambertian emission, in the pixels of the source image, we have a calibrated (truth) average value $DN_{truth}$ given by(12)$$DN_{truth}=k\tau {\langle L_{truth}\left(\lambda \right)\rangle }_{\Delta \lambda }+DN_{dark\;truth}$$
with $DN_{dark\;truth}$ being the value read with the generic display pixel off (dark measurement). ${\langle L_{truth}\left(\lambda \right)\rangle }_{\Delta \lambda }$ is numerically computed for the DALSA spectral response using its detector spectral response $S_{D}(\lambda )$ (from datasheet) convolved with the (known) spectrum $L_{truth}\left(\lambda \right)$ of the calibrated source. Knowing ${\langle L_{truth}\left(\lambda \right)\rangle }_{\Delta \lambda }$, τ, $DN_{truth}$, and $DN_{dark\;truth}$, the calibration constant k can be determined and used to characterize the spectrum of the MINISTAR display.

Each pixel of the MINISTAR RGB OLED emits radiance in a specific color channel (in our case, the red channel was selected for simulation). Even if the spectral profile of the OLED display is known from its datasheet, its amplitude has to be radiometrically characterized. The response of DALSA observing the MINISTAR display can be determined following the same procedure.

By setting the MINISTAR display to emit spatially uniform brightness, the average digital number signal $DN_{MS}$ for a generic MINISTAR pixel is given by(13)$$DN_{MS}=k\tau {\langle R\;r_{MS}\rangle }_{\Delta \lambda }+{DN_{dark}}_{MS}$$
with ${DN_{dark}}_{MS}$ being the dark signal measured with the display showing a black image, $r_{MS}(\lambda )$ the spectral response of the red channel (as provided in the datasheet) normalized to unity, and R its constant scale factor representing the amplitude (scale factor) of the normalized spectral response $r_{MS}(\lambda )$. As in the case of the calibrated source, the spectrally averaged radiance measured by the DALSA detector is(14)$${\langle R\;r_{MS}(\lambda )\rangle }_{\Delta \lambda }=R\int_{\Delta \lambda }S_{D}\left(\lambda \right)\;r_{MS}\left(\lambda \right)\;d\lambda =R{\langle r_{MS}(\lambda )\rangle }_{\Delta \lambda }=\frac{DN_{MS}-{DN_{dark}}_{MS}}{k\;\tau }$$

By numerically computing ${\langle r_{MS}(\lambda )\rangle }_{\Delta \lambda }=\int_{\Delta \lambda }S_{D}\left(\lambda \right)r_{MS}\left(\lambda \right)d\lambda$, the constant R can be estimated as(15)$$R=\frac{DN_{MS}-{DN_{dark}}_{MS}}{k\;\tau \int_{\Delta \lambda }S_{D}\left(\lambda \right)\;r_{MS}\left(\lambda \right)\;d\lambda }$$

Figure 10 shows the spectral radiance of the red channel of the MINISTAR display ${\langle R\;r_{MS}\rangle }_{\Delta \lambda }$, the calibrated source HL-3P-INT-CAL spectrum, and the normalized spectral response of the DALSA 1M60 detector. As detailed in [10], a series of tests was also performed to assess the linearity of the MINISTAR display with increasing brightness levels (electronically set).

Additional tests evaluated the system’s ability to reproduce the magnitude of stars listed in the HIPPARCOS catalog after calibration. Specifically, Figure 11 compares the magnitudes of stars in the Cassiopeia constellation simulated by the MINISTAR display with the corresponding visual magnitudes listed in the HIPPARCOS catalog.

### 5.2. STARLITE Performance Evaluation and Future Development

STARLITE builds on the heritage of the earlier MINISTAR prototype, which reached TRL 5 through development funded by the Tuscany Region under the 2014–2020 Regional Operational Program (FESR). The STARLITE project continues this line of development, explicitly targeting the resolution of technical limitations observed in MINISTAR in order to deliver a commercially viable, flight-representative validation tool. The system is intended to achieve a TRL of at least 7, with form factor and mass optimized for direct integration onto satellite-mounted star trackers.

Although compact star field simulators are currently scarce on the market, demand from international space agencies is increasing, particularly for systems capable of performing functional testing of star trackers in their final integrated satellite configuration. This need reflects a growing emphasis on near-operational test scenarios—what is effectively “on-orbit-like” validation. The STARLITE system is being developed precisely to meet this emerging requirement by providing a versatile, lightweight, and high-performance validation platform.

The feasibility of such a miniaturized simulator has only recently been enabled by advancements in electro-optical components and integration technologies. The presence of a globally recognized star tracker manufacturer in Tuscany, along with the industrial involvement of regional small and medium-sized enterprises (SMEs), provides a unique synergy. This collaboration is expected to strengthen the regional space technology cluster, promote economic growth, and generate new employment opportunities.

Key improvements in the STARLITE design over its predecessor include the following:Enhanced microdisplay interface electronics supporting higher resolution;Collimating optics based on Commercial Off-The-Shelf (COTS) components, enabling significant simplification of mechanical structure and the focusing/alignment system;An upgraded control software suite for precise and flexible system operation.

These developments will enable STARLITE to reach TRL ≥ 7, establishing it as a reliable, compact, and fully integrated star field simulator for advanced validation of spaceborne attitude determination sensors.

#### 5.2.1. Enhanced Calibration Procedure Planned for STARLITE

An enhanced calibration procedure is planned for STARLITE, with particular emphasis on improving geometric calibration. Building on the geometric characterization outlined in Section 5.1.1, the procedure incorporates corrections for image distortion introduced by the calibration camera. Unlike the MINISTAR calibration process, which assumed the camera image to be ideal (i.e., free of distortion), the revised procedure explicitly accounts for uncertainties introduced by the camera’s optical system. A key novelty of the updated approach is the incorporation of a dedicated camera geometric calibration step into the procedure. This refinement enables improved precision in the simulated star positioning. The proposed calibration process includes the following steps:Camera geometric calibration: In this phase, the camera, focused at infinity, captures images of a regular pattern. The geometry of the setup must be precisely known so that, given the image distance and target dimensions, the viewing angles corresponding to each pixel can be determined relative to the imaging axis. This step allows for the intrinsic calibration of the camera, including correction of any residual optical distortion. For MINISTAR, a distortion-free camera image was just assumed.STARLITE geometric calibration: A grid of regularly spaced points is displayed on the STARLITE screen and imaged using the previously calibrated camera. The distorted points are then mapped as in Section 5.1.1, and, by mapping inversion, STARLITE optical system’s geometric distortion can be minimized.

This two-step process is expected to significantly improve the angular accuracy of the STARLITE system compared to its predecessor, MINISTAR, enabling more precise mapping between display coordinates and viewing directions by removing the camera intrinsic distortion.

#### 5.2.2. Display Luminance Conversion to Apparent Magnitude

Based on the specifications of the display (SeeYA SY103WAM01 2560 × 2560 × 3 RGB), an estimate of the device’s radiometric output is performed for the purpose of simulating stellar magnitudes.

Since the conversion between photometric and radiometric quantities depends on the source spectrum, a simplification is made by assuming a conversion at 555 nm (1 W = 683 lm), which corresponds to the peak of the photopic luminous efficiency curve [15]. This assumption is subject to variation depending on the actual spectral emission of the device; accurate calibration will be carried out during the device verification phase. The conversion at 555 nm should be interpreted as a preliminary estimate of the luminous output of a single pixel at the device’s maximum luminance (1800 cd·m^−2^ for the display used for STARLITE).

The display luminance (in cd m^−2^) was converted into radiance (W m^−2^ sr^−1^) and integrated over the solid angle subtended by an individual pixel. Given the focal length f of the collimator, the instantaneous field of view (IFOV), or solid angle subtended by a pixel with pitch p, is given by $IFOV=\frac{p^{2}}{f^{2}}$.

Using the apparent bolometric magnitude of the star Vega as the zero-point $m_{ref}$ for apparent magnitude, the maximum apparent magnitude m emitted by a single pixel, observed through a 35 mm collimator, is estimated to be approximately m≈−1.6.

This same result can also be obtained by setting $m_{ref}$ equal to the apparent magnitude of the Sun and using the corresponding irradiance (in W m^−2^) listed in Table 4.

#### 5.2.3. Vignetting Problem

The STARLITE device produces an image focused at optical infinity and composed of parallel light rays. Consequently, these rays diverge after exiting the STARLITE exit pupil as a function of distance, a purely geometric effect illustrated in Figure 12. As the distance between the STARLITE exit pupil and the star tracker entrance pupil under test increases, an increasing fraction of rays falls outside the star tracker entrance pupil, resulting in vignetting. This leads to a progressive decrease in image brightness towards the periphery.

The maximum distance to avoid vignetting can be defined geometrically. Typically, the distance between the collimator lens and the star tracker optical head is on the order of a few mm. To address this, STARLITE includes a flange (visible in Figure 13) designed to be inserted into the star tracker baffle, positioning the collimator as close as possible to the first optical surface of the star tracker under test.

A future measurement campaign is scheduled for STARLITE, aimed at quantitatively assessing the extent and impact of vignetting in commercially available star trackers.

As detailed in the preceding sections, the focal length of the collimation system is given by Equation (9). It is preferable to select the widest possible FOV compatible with the optical requirements to ensure the test system accommodates most commercially available star trackers, which typically have an FOV around 25°. Vignetting occurs when the collimator and the star tracker imaging system cannot be approximated as ideal thin lenses. In multi-element optical systems, if the equivalent thin lens lies within the physical length of the collimator and inside the star tracker’s enclosure, the distance between these lenses may become large enough to induce vignetting, thus limiting the effective simulated FOV. Figure 14 illustrates this effect through a comparison between images obtained by projecting a test image into a camera at increasing distance: vignetting can be observed.

#### 5.2.4. Comparison with Existing Systems

Table 5 presents a comparative summary of the current capabilities of STARLITE against other notable star tracker test systems, including STOS by Airbus Space Equipment [7] and OSI by Jena-Optronik [8]. STARLITE offers key advantages in terms of higher display resolution and the ability to simulate a substantially larger FOV. While the nominal FOV exceeds 29° (corresponding to an 18.434 mm image height with a 35 mm focal length), ongoing validation is being conducted to confirm the absence of vignetting when interfaced with various star tracker models. As a precaution, the values reported in Table 5 reflect conservative estimates.

A distinctive feature of STARLITE is its capacity to perform parallel simulations for multiple star tracker optical heads (in the predecessor MINISTAR, this number was limited to three heads). These can be positioned with independent orientations within the same simulation environment, enabling a thorough evaluation of multiple sensors operating concurrently on a shared satellite platform.

In addition, STARLITE supports the use of both standard (Hipparcos) and custom star catalogs, including synthetic objects. This functionality is particularly useful for calibration and alignment verification procedures, where predefined star patterns (such as angularly equidistant stars with constant or systematically varying magnitudes) are often required. These can be easily implemented through user-defined catalogs.

The system’s graphical user interface (GUI) further enhances its operational flexibility by providing seamless integration between the simulation environment and the user’s star tracker software model. The GUI enables efficient management of star patterns, execution of alignment procedures, and injection of controlled disturbances (such as stray light or particle impacts) into the simulation. These capabilities contribute to a highly realistic and versatile test environment that is well suited for validating modern star tracker systems under complex operational scenarios.

## 6. Discussion

This paper has provided an overview of the development process and architecture of the MINISTAR prototype and its evolution into the STARLITE OGSE system, designed for the testing and validation of star trackers. The development of the early prototype MINISTAR and the STARLITE project is being carried out through a collaborative effort involving several Tuscan industrial and research entities: SAITEC S.R.L. (project leader), GESTIONE SILO S.R.L., PROMEL COSTRUZIONI S.R.L., and the Institute of Applied Physics “Nello Carrara” of the Italian National Research Council (IFAC-CNR).

STARLITE represents a substantial advancement over its predecessor MINISTAR, benefiting from improved system design, component upgrades, and a more refined integration strategy. The experience gained from the MINISTAR phase has been instrumental in addressing critical issues and enabling STARLITE to reach a TRL of at least 7. Although development activities are still ongoing, the current work outlines the foundational steps in building the STARLITE system. Significant improvements have been implemented in the simulation model, with a focus on the accurate reproduction of celestial scenes, enhanced optical precision, and high radiometric fidelity, which are crucial features for effective star tracker validation. Hardware upgrades include the adoption of a high-resolution display and a commercial-grade fixed-focal-length photographic objective with a standard C-mount interface, functioning as a collimator. This setup facilitates straightforward replacement of the optics, allowing the simulation of different FOVs by simply exchanging objectives with varying focal lengths.

An ongoing detailed analysis of vignetting between the collimator and the star tracker’s optical head is being conducted to identify and characterize the system’s limitations in optical coupling. Additionally, the adoption of a high-resolution display for star rendering has significantly improved luminance compared to MINISTAR. The enhanced display resolution provides finer angular sampling, enabling accurate simulation of stellar positions across larger FOVs. The adoption of a square form factor enables more efficient use of the display area (in contrast, MINISTAR left portions of the display unused when simulating a square FOV). These advancements substantially improve the system’s capability to test and validate contemporary star trackers under more demanding and realistic operational conditions.

Another significant improvement concerns the redesign of the interface flange connecting STARLITE to the star tracker’s baffle. This redesign addresses the mechanical and alignment issues observed during the MINISTAR development phase.

These collective improvements position STARLITE as a robust and competitive testing platform that is well aligned with the requirements of future industrialization and market deployment. The use of commercial off-the-shelf (COTS) components also contributes to reducing overall system complexity.

Looking ahead, the next development phases will include a comprehensive calibration campaign to characterize STARLITE’s simulation fidelity and determine the operational limits of the integrated commercial optical components. Further refinements will be made to the mechanical housing to ensure adaptability to the form factors of most commercial star trackers, supporting the system’s transition to a market-ready product.

The STARLITE system demonstrates strong commercial viability due to its compact design, ease of use, and high-performance capabilities, making it well suited for both laboratory testing and potential integration on satellite platforms. As the demand for high-resolution spaceborne instruments continues to grow, particularly in Earth observation and scientific missions, there is an increasing need for precise attitude determination. This, in turn, requires accurate pre-launch testing of star trackers and other attitude sensors. STARLITE addresses this need by offering a versatile and scalable solution capable of simulating wide fields of view with high angular resolution. With its modular design, careful planning, and advanced technical features, STARLITE lays a solid foundation for a scalable and reliable OGSE solution in the aerospace sector.

## Figures and Tables

**Figure 1 sensors-25-05360-f001:**
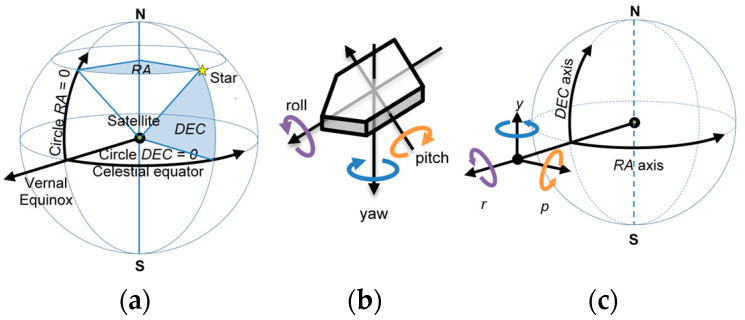
(**a**) Celestial coordinate system and declination and right ascension angles DEC and RA. RA angle is measured along the celestial parallel starting at the Vernal Equinox (one of the intersection points between the ecliptic and the celestial equator circles) and ranging from 0 to 360°. The DEC angle measures the angular distance from the celestial equator, ranging from −90° to +90°. (**b**) Tait–Bryan angles describing the roll, pitch, and yaw angles around, respectively, the x, y, and z axes. (**c**) The roll, pitch, and yaw angles and the x, y, and z axes in the Celestial Sphere reference frame.

**Figure 2 sensors-25-05360-f002:**
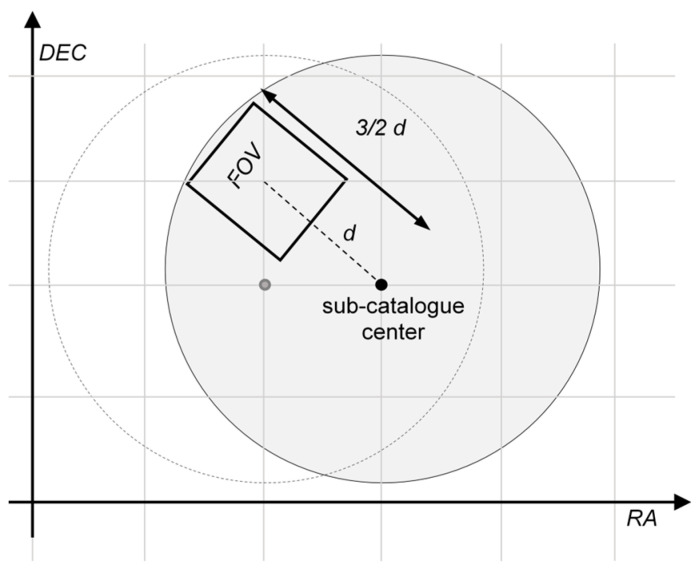
The star catalog is divided into multiple spatially overlapping sub-catalogs. The angular width of each region is large enough to contain the FOV of the simulated scene. The direction of view is used to select the sub-catalog of interest, so only stars entering the FOV of the simulated scene are selected for applying the inverse rotation matrix, bringing them into the star tracker’s optical head reference frame.

**Figure 3 sensors-25-05360-f003:**
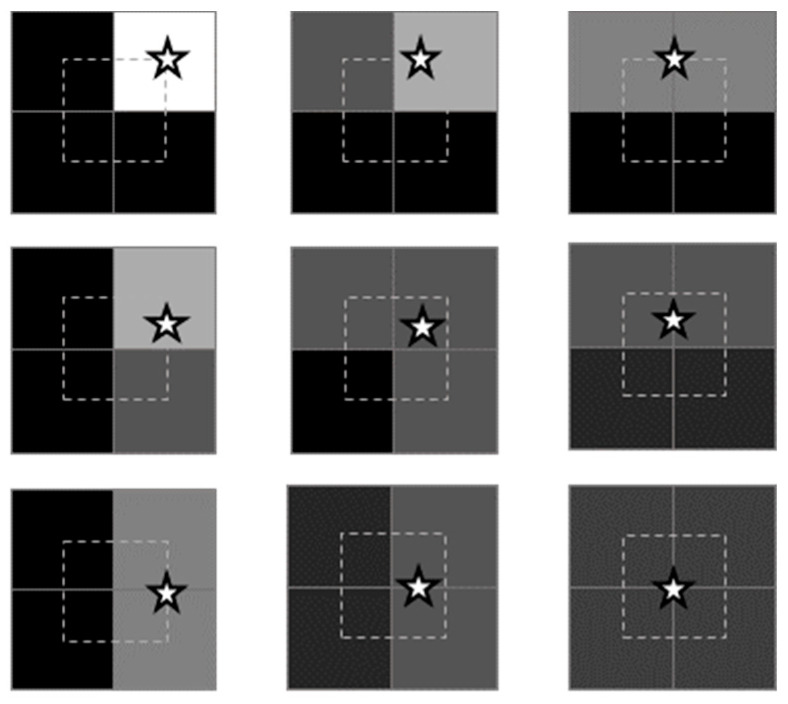
The 2 × 2 pixel matrices used for representing the position of a star with sub-pixel precision. The gray level of each pixel represents its luminance value 0 (black) to 255 (white). The star position lies in the barycenter of the 2 × 2 pixel distribution, and the sum of the pixel values over the matrix gives the luminance value associated with the star. The use of a 2 × 2 pixel matrix allows for varying the barycenter position both in the horizontal (matrices left to right) and I vertical directions (matrices top to bottom) in the range of ±1/2 pixel (segmented square) around the 2 × 2 matrix center. Note that only variations in the upper rightmost quadrant are shown due to the symmetry of the problem.

**Figure 4 sensors-25-05360-f004:**
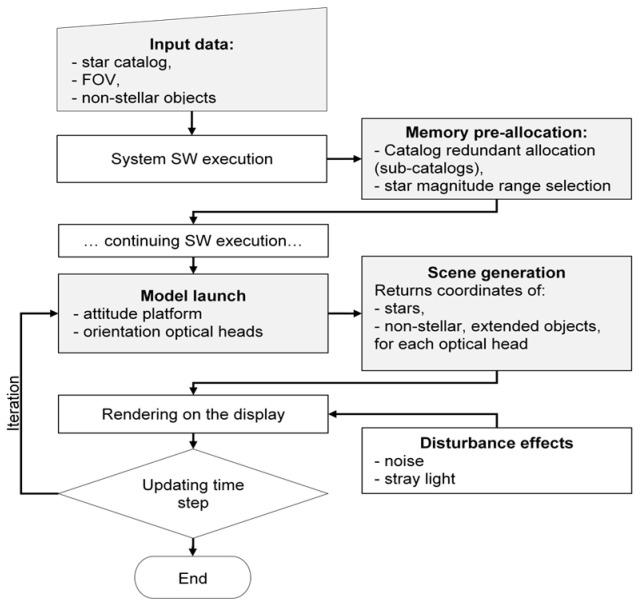
Logic flux of the mathematical model providing the representation of the observed scene by the generic star tracker’s optical head. Pre-allocation represents the “slow” computation phase. Such architecture allows fast (real-time) computation on a limited set of celestial bodies (both stars and non-stellar objects) at the simulation phase. Disturbance effects are input into the simulation before the rendering of the simulated scene. Model-related blocks are highlighted in grey, and general system software execution is in white.

**Figure 5 sensors-25-05360-f005:**
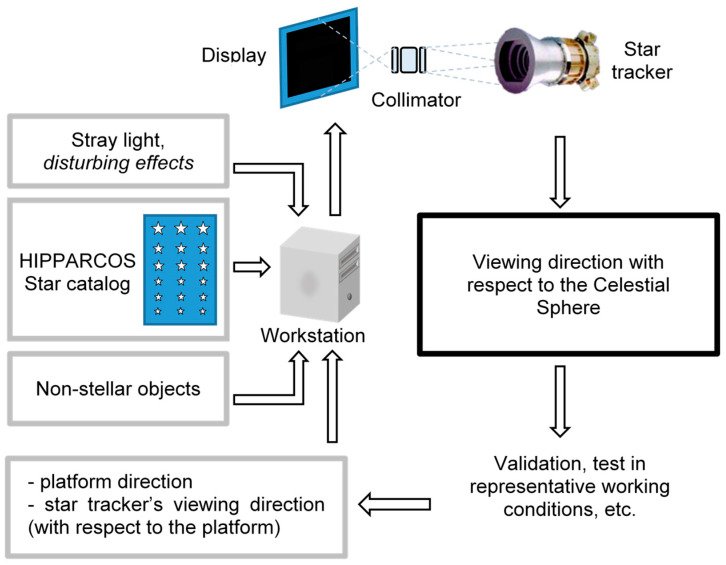
MINISTAR and STARLITE star tracker validation system architecture. Input data (outlined in gray) are processed by a workstation, providing the dynamic, real-time rendering of the scene on the instrument’s display and projected after collimation into the star tracker optical head under test. Star tracker output data (outlined in black) can be analyzed, allowing checks in both open and closed-loop configurations.

**Figure 6 sensors-25-05360-f006:**
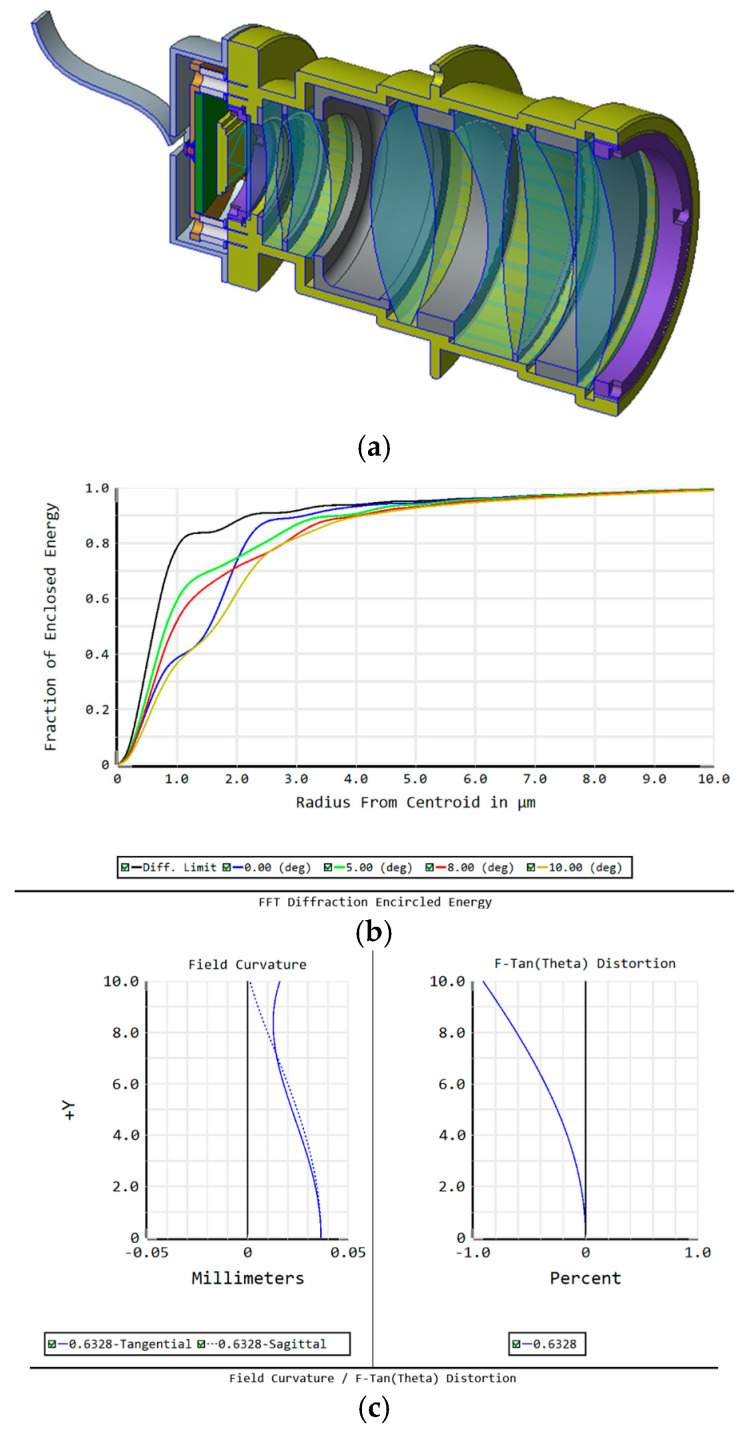
(**a**) MINISTAR optics and mechanical mounting. (**b**) Encircled energy for the MINISTAR optical system (generated by Zemax OpticStudio). (**c**) Optical system field curvature and distortion (generated by Zemax OpticStudio).

**Figure 7 sensors-25-05360-f007:**
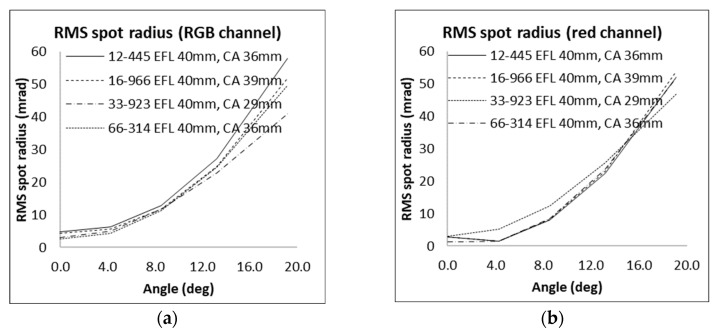
Edmund lenses RMS spot radius (in mrad, afocal mode) for different angles for the case of (**a**) an RGB source; and (**b**) for the red channel only. The resulting angular resolution is inferior to the theoretical limit imposed by the display pixel size, primarily due to lens-induced aberrations.

**Figure 8 sensors-25-05360-f008:**
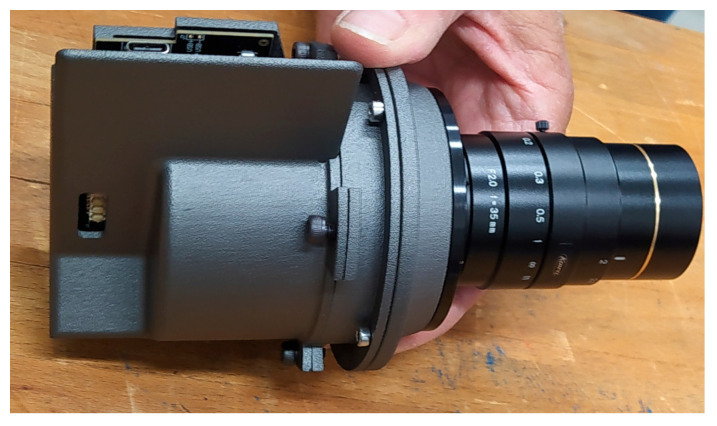
One of the STARLITE current shelves realized with 3D printing, mounting the Kowa 35 mm objective LM35XC by Kowa Optimed, Deutschland GmbH (Duesseldorf, Germany). STARLITE supports standard C-mount optics, enabling flexible FOV adjustment via interchangeable objectives.

**Figure 9 sensors-25-05360-f009:**
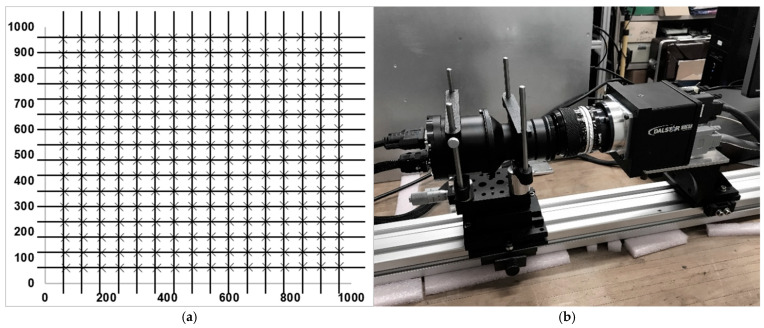
(**a**) The diagram shows the MINISTAR optics-distorted pixels (crosses) versus the ideal (regularly spaced) coordinates (black grid). By knowing the regularly spaced coordinates in the input, the optics-induced distortion can be mapped and corrected. (**b**) Distortion mapping experimental setup. The MINISTAR prototype (on the left) projects the point grid on the camera (right).

**Figure 10 sensors-25-05360-f010:**
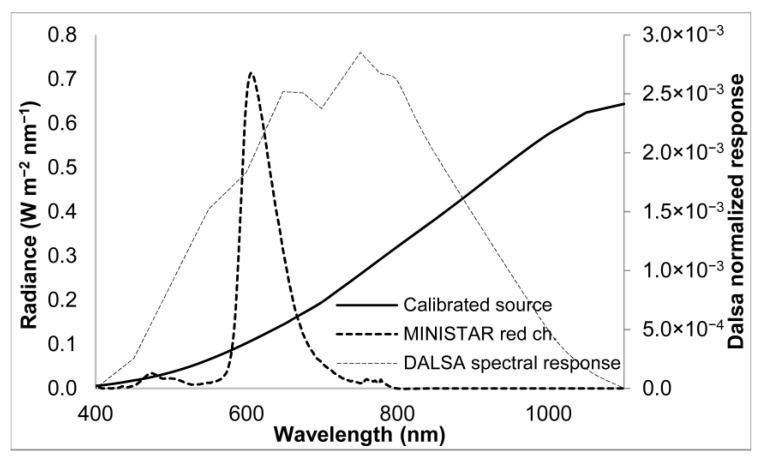
Spectral radiance of the red MINISTAR display channel ${\langle R\;r_{MS}\rangle }_{\Delta \lambda }$ (dotted), the spectrum of the calibrated source HL-3P-INT-CAL with Lambertian diffuser (continuous), and the normalized spectral response of the DALSA 1M60 detector (dashed line).

**Figure 11 sensors-25-05360-f011:**
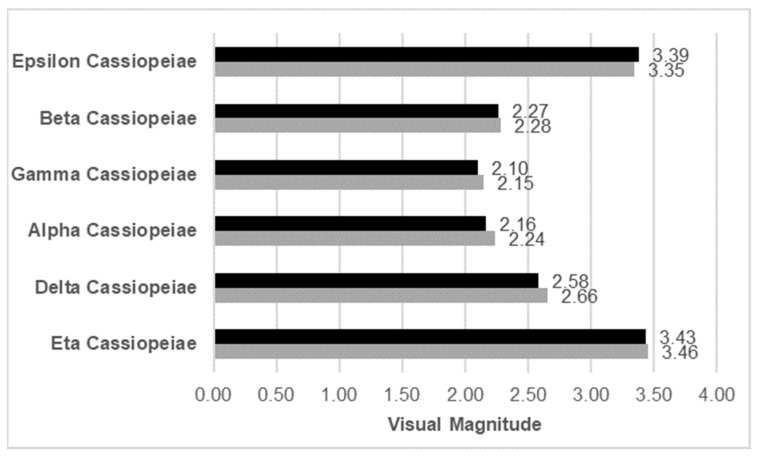
Visual magnitude comparison for a set of selected stars: the magnitude values simulated by the MINISTAR (after calibration) are shown in grey, and the corresponding values of apparent magnitude from the HIPPARCOS star catalog are shown in black.

**Figure 12 sensors-25-05360-f012:**
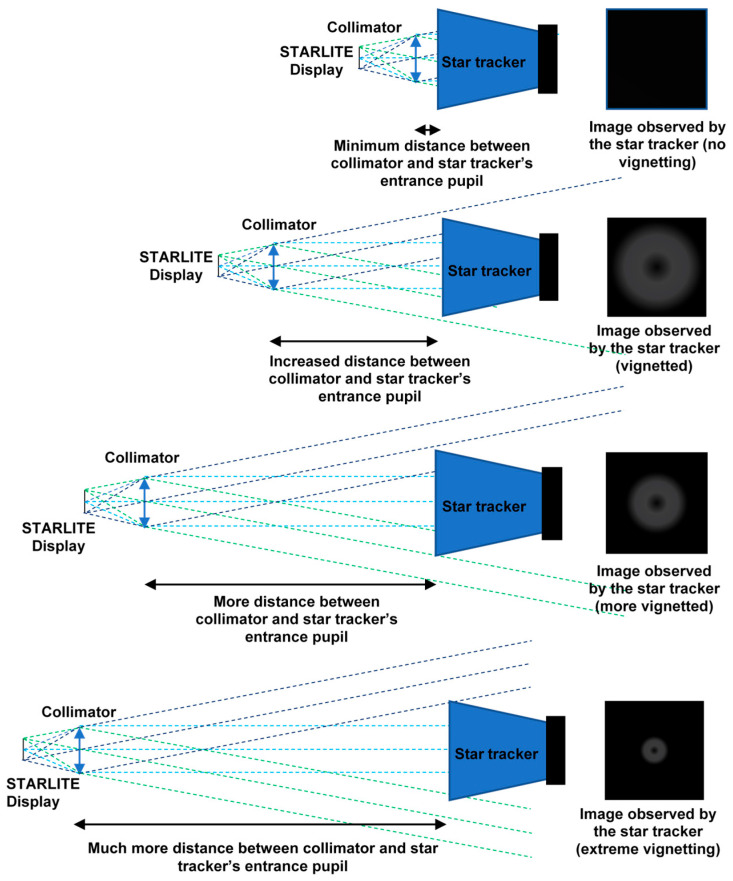
Vignetting problem: As the separation between the STARLITE exit pupil and the entrance pupil of the star tracker under test increases, a growing portion of the emitted rays falls outside the acceptance aperture of the star tracker, leading to increased vignetting.

**Figure 13 sensors-25-05360-f013:**
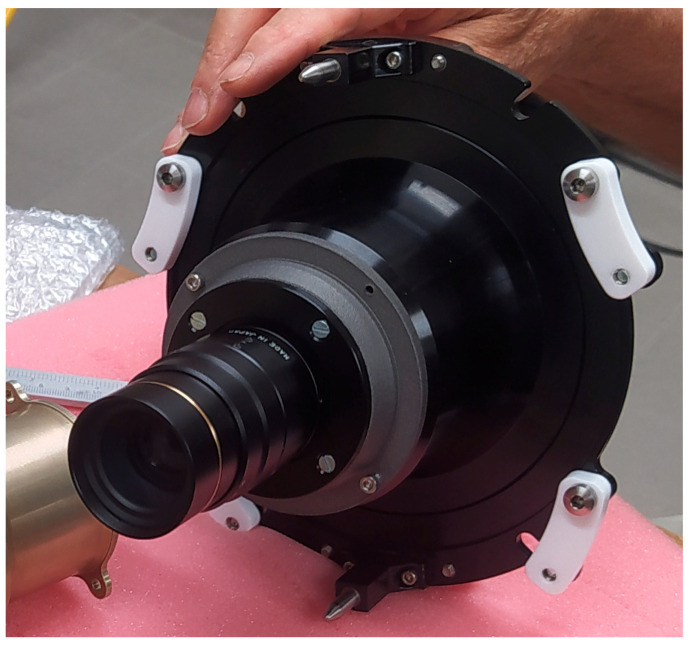
STARLITE flange for mounting on baffle.

**Figure 14 sensors-25-05360-f014:**
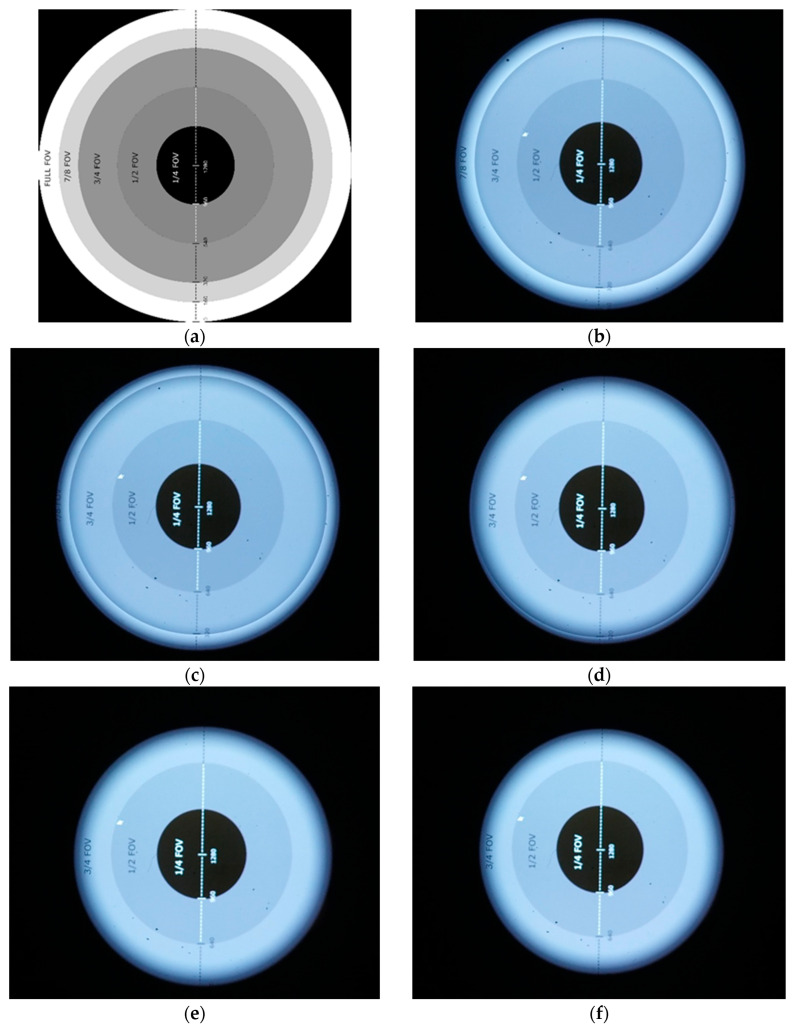
Vignetting with increasing distance between camera (simulating a star tracker) and STARLITE collimator. (**a**) Test image. (**b**) Minimal vignetting. (**c**–**f**) Increasing vignetting with distance between STARLITE and the camera.

**Table 1 sensors-25-05360-t001:** Key parameters for the selection and sizing of optoelectronic components in the MINISTAR prototype.

Parameter	Description
Input	Star catalog (positions and magnitudes) Support for custom catalogs Platform position (celestial coordinates) Platform rotation rate Relative position of the *n*-th head with respect to the platform Coordinates (Celestial Sphere) and magnitudes for simulating large objects (planets, Sun, Moon, Earth, etc.) Noise effects (e.g., cosmic rays) Straylight profile (customizable)
Field of View (degrees)	20°
Dynamic Scene Frame Rate	85 frame/s
Exit Pupil Diameter	35 mm
Dimensions	175 mm (⌀) × 150 mm
Weight	~1 kg
Alignment Accuracy	0.001°
Angular Accuracy for Star Positioning	0.005°
Star Magnitude Range	4.5 ± 0.2 mag
Alignment Software Calibration	Foreseen (supported by SW model)
Radiometric Software Calibration	Foreseen (supported by SW model)
Use in Thermal Vacuum Environment	Not planned

**Table 2 sensors-25-05360-t002:** Target design parameters for the STARLITE system.

Parameter	Description
Input	Star catalog (positions and magnitudes) Support for custom catalogs and calibration patterns of stars for calibration/alignment Platform position (celestial coordinates) Platform rotation rate Relative position of the *n*-th optical head with respect to the platform Coordinates (Celestial Sphere) and magnitudes for simulating large objects (planets, Sun, Moon, Earth, etc.), customizable objects (user-defined) Noise effects (e.g., cosmic rays), customizable pattern and rate Straylight profile (customizable, supporting straylight image as input)
Field of View (degrees)	25° to 30° diagonal (field diameter > 20°)
Dynamic Scene Frame Rate	60 to 90 Hz
Exit Pupil Diameter	Actually ≤43 mm (for requirements on minimum star tracker’s baffle internal diameter)
Physical Dimensions	Comparable to those of MINISTAR
Weight	≤MINISTAR
Alignment Accuracy	≤0.001° (to be determined)
Angular Accuracy for Star Positioning	≤0.005° (to be determined)
Star Magnitude Range	4.5 ± 0.2 mag
Alignment Calibration via Software	Supported
Radiometric Calibration via Software	Supported
Operation in Thermal Vacuum	Not currently foreseen

**Table 3 sensors-25-05360-t003:** The analysis of Edmund Optics components focused on achromatic and aspheric lenses with a focal length of 40 mm, selected to provide an FOV comparable to that of the MINISTAR prototype.

Focal	Lens Type	Model N.	Producer
40 mm	Aspheric Lens	#16-966	Edmund
40 mm	Aspheric Lens	#12-445	Edmund
40 mm	Achromatic Lens	#33-923	Edmund
40 mm	Aspheric Lens	#66-314	Edmund

**Table 4 sensors-25-05360-t004:** Display luminance conversion to apparent magnitude. The Vega star radiative luminosity at 10 parsec (W/m^2^) has been converted into nominal display lumens using the reported conversion factor.

Display nominal luminance (cd m^−2^ = lm m^−2^ sr^−1^)	1800
lumen to watt conversion factor (w lm^−1^)	0.001464129
watt to lumen conversion factor (lm w^−1^)	683
Number of pixels on display side	2500
Display side (mm)	18.432
Collimator focal length (mm)	35
Vega 0 point IAU [16] radiative luminosity (W)	3.01280 × 10^28^
parsec to m conversion factor (m)	3.08568 × 10^16^
Vega 0 point apparent (at 10 parsec) bolometric magnitude (W/m^2^)	2.51802 × 10^−8^
0 point apparent bolometric magnitude (Vega)	0
Nominal total solar irradiance (W m^−2^)	1361
Sun apparent magnitude	−26.832
pixel pitch (mm)	0.0073728
IFOV solid angle (sr)	4.4374 × 10^−8^
Radiance (@555 nm) (W m^−2^ sr^−1^)	2.635431918
Irradiance (@555 nm) (W m^−2^)	1.16945 × 10^−7^
1 pixel Bolometric magnitude	−1.667303609
1 pixel Bolometric magnitude at 20% of luminance	0.080121402
4 pixels Bolometric magnitude at 20% of luminance	−1.425028576
Display nominal luminance (cd m^−2^ = lm m^−2^ sr^−1^)	1800
lumen to watt conversion factor (w lm^−1^)	0.001464129

**Table 5 sensors-25-05360-t005:** Comparison of STARLITE main features with STOS (Airbus Space Equipment) and OSI (Jena-Optronik) systems.

	STARLITE	STOS	OSI
Close loop stimulation	Yes	Yes	Yes
Display dimensions	2560 × 2560 pixels	1280 × 1024 pixels	-
FOV	≥25° (±12.5°)	25° (±12.5°)	>20°
Frame rate for dynamic scene	Up to 90 Hz	255 Hz monochrome	≥60 Hz
Device dimensions	175 mm (⌀) × 150 mm	200 mm (⌀) × 350 mm	77 mm (⌀) × 180 mm
Weight	<1 kg	<2 kg	<1 kg
Alignment error	0.001°	0.001°	-
Attachment	star tracker baffle	star tracker baffle	star tracker baffle
Single star accuracy	0.005°	0.005°	0.0075°
Star catalog	Hipparcos/custom	Hipparcos	Hipparcos
Star magnitude range	4.5	4.5	4.5
Star diameter	<0.05°	<0.13°	<0.1°
Large objects	Yes	Yes	Yes
Protons simulation	Yes	Yes	Yes
Stray light simulation	Yes	Yes	Yes

## Data Availability

Available data are contained within the article.

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
