# Peer review of "MINISTAR to STARLITE: Evolution of a Miniaturized Prototype for Testing Attitude Sensors"

_sensors, 2025, doi:10.3390/s25175360_

Round 1
Reviewer 1 Report
Comments and Suggestions for Authors
The article presents the engineering design and performance evaluation of a device for testing star trackers. The authors present the design of the device, the concept of operations, an evaluation of its accuracy, and overall performance.
The article is presented mostly as an engineering data sheet for the device, as opposed to the results of a research project. The introduction eludes to this device providing benefits that other devices do not exude, such as being miniature and able to be installed directly on the instrument baffle.
Examples of other similar devices are provided, including Airbus's STOS and the Jena-Optronik ASTRO APS. While it's true that the STOS cannot be mounted onto the baffle of a star tracker, the ASTRO APS most certainly can.
The paper provides no comparison of any parameters to the existing state of the art, including mass, dimensions, or performance. It is thus impossible to see what the tangible research contributions are in the paper. As mentioned earlier, it reads more like an advertisement or data sheet for this single device. If I, as a satellite systems integrator was interested in using this device for my satellite, I would find this a very useful document, but as a research paper, I struggle to see the contribution.
In order for this to be acceptable in a research journal, the scientific contributions must be clearly stated and demonstrated in the context of existing solutions. Right now, that is not present.
Author Response
We sincerely thank the reviewer for their careful and critical assessment of our manuscript. We appreciate your detailed feedback and the opportunity to improve the clarity and positioning of our work.
We understand the concern regarding the perception of the manuscript as more of an engineering report or product data sheet than a research paper. In response, we have revised the manuscript significantly to better highlight the scientific contributions, particularly in the context of the current state of the art. In particular, we have made the following key revisions:
- Introduction was modified in order to introduce the improvement of STARLITE with respect to MINISTAR.
- A more comprehensive discussion highlighting the differences between MINISTAR and STARLITE has been included in both the Introduction and, in greater detail, in Section 4.1. Furthermore, section 4.2 has been modified accordingly.
- Additional details regarding the geometric characterization have been added in Section 5.1.1.
- A new Section 5.2.1 has been introduced, describing the calibration procedures in major details.
- A comparative analysis with existing systems is now presented in Section 5.2.4, and the key performance features are summarized in the newly added Table 5.
- A brief description of the GUI’s usability features has been incorporated into Section 5.2.4.
- Finally, a discussion on the potential commercial viability of the STARLITE system has been included in the revised manuscript under the Discussion section.
- Minor editing has been performed. All modifications are highlighted as revision marks in the revised manuscript.
We thank you once again for your constructive review and support.
With best regards,
The Authors

Reviewer 2 Report
Comments and Suggestions for Authors
This article describes a compact electro-optical system designed to create artificial fields of stars in visible motion for realistic ground-based testing of star sensors. The operating principles of the system and the capabilities of the developed prototype are described: modeling of stars and other celestial bodies, user-defined objects and disturbance sources.
The topic of the work is relevant, since the developed device will allow more accurate testing of star sensors.
It is possible to note the compactness of the developed device in comparison with known prototypes.
The article is prepared at a high level of scientific publication quality.
The conclusions are consistent with the evidence and arguments presented and answer the main research question posed.
The references to sources are quite complete and allow one to assess the significance of the work done by the authors.
The presented figures and tables are of good quality and sufficiently explain the results of the study.
Author Response
We would like to sincerely thank the reviewer for their thoughtful and positive evaluation of our manuscript. We greatly appreciate your recognition of the scientific relevance of our work and are especially grateful for your encouraging comments. Your feedback reinforces the value of our research and motivates us to continue improving and expanding the capabilities of the STARLITE system.
In response to the observations raised by the other reviewers, we have made several revisions to enhance the manuscript:
- Introduction was modified in order to introduce the improvement of STARLITE with respect to MINISTAR.
- A more comprehensive discussion highlighting the differences between MINISTAR and STARLITE has been included in both the Introduction and, in greater detail, in Section 4.1. Furthermore, section 4.2 has been modified accordingly.
- Additional details regarding the geometric characterization have been added in Section 5.1.1.
- A new Section 5.2.1 has been introduced, describing the calibration procedures in major details.
- A comparative analysis with existing systems is now presented in Section 5.2.4, and the key performance features are summarized in the newly added Table 5.
- A brief description of the GUI’s usability features has been incorporated into Section 5.2.4.
- Finally, a discussion on the potential commercial viability of the STARLITE system has been included in the revised manuscript under the Discussion section.
- Minor editing has been performed. All modifications are highlighted as revision marks in the revised manuscript.
We thank you once again for your constructive review and support.
With kind regards,
The Authors

Reviewer 3 Report
Comments and Suggestions for Authors
The paper "MINISTAR to STARLITE: evolution of a miniaturized prototype for testing attitude sensors" presents an overview of the development and capabilities of the MINISTAR system and its evolution into the STARLITE system. It shows the importance of star trackers in satellite attitude determination and the need for effective ground-based testing solutions.
I see this paper as a technological paper, describing equipment that are very important in aerospace engineering. It shows detailed descriptions of the design criteria, operational principles, and performance evaluations of both systems.
I have the following comments/questions:
- The paper should make a better discussion on the limitations observed in the MINISTAR prototype and how they are addressed in STARLITE.
- Can you give some more detail on the calibration methods used for both geometric and radiometric parameters? How do these methods give accuracy of star field simulations?
- What are the key performance metrics for STARLITE, and how do they compare to those of existing systems in the market?
- How does the graphical user interface facilitate ease of use and test configuration?
- Do you believe in commercial viability of the STARLITE system?
If the journal is looking for papers of this type, I see this one as a good candidate for publication. It is well written and the figures have good quality.
Author Response
We sincerely thank you for your valuable feedback and thoughtful comments on our manuscript. We have carefully considered each point and made corresponding revisions to improve the quality and clarity of the paper. Please find below our detailed responses, organized by comment:
Q1. The paper should make a better discussion on the limitations observed in the MINISTAR prototype and how they are addressed in STARLITE.
A1. We have included a more comprehensive discussion highlighting the differences between MINISTAR and STARLITE both in the introduction and, in particular, in Section 4.1. This section outlines the key improvements introduced in STARLITE to overcome the limitations observed in the MINISTAR prototype.
Q2. Can you give some more detail on the calibration methods used for both geometric and radiometric parameters? How do these methods give accuracy of star field simulations?
A2. In response to this request, we have added more details in Section 5.1.1 and added a new Section 5.2.1, which describes the proposed calibration procedures for STARLITE, including both geometric and radiometric aspects. This addition clarifies how these methods contribute to improving the accuracy of the simulated star fields and describes the differences between the calibration procedures of MINISTAR and STARLITE.
Q3. What are the key performance metrics for STARLITE, and how do they compare to those of existing systems in the market?
A3. A description of STARLITE’s key performance metrics and a comparative analysis with existing systems are now provided in Section 5.2.4. For clarity, Table 5 has been added, showing their main performance features.
Q4. How does the graphical user interface facilitate ease of use and test configuration?
A4. Although the GUI is still under final definition, we have added a brief description in Section 5.2.4 outlining its usability features and its intended role in facilitating test configuration and user interaction.
Q5. Do you believe in commercial viability of the STARLITE system?
A5. A discussion on the potential commercial viability of the STARLITE system has been included in the revised manuscript under the Discussion section. We highlight factors such as modular design, COTS integration, and system versatility that support its commercial potential.
Minor editing has been performed. All modifications are highlighted as revision marks in the revised manuscript.
We are grateful for the reviewer’s constructive input, which has significantly contributed to improving the manuscript. We hope that the revised version satisfactorily addresses all comments and concerns.
With kind regards,
The Authors

Round 2
Reviewer 3 Report
Comments and Suggestions for Authors
I thanks the authors for taking into acccount my comments. I consider the paper is ready for publication in its current format.